# Collaborative Determination Method of Metro Train Plan Adjustment and Passenger Flow Control under the Impact of COVID-19

**Fuquan Pan** [1,*], **Jingshuang Li** [1], **Hailiang Tang** [2], **Changxi Ma** [3], **Lixia Zhang** [1] and **Xiaoxia Yang** [1]

1 School of Civil Engineering, Qingdao University of Technology, Qingdao 266520, China
2 Qingdao Metro Operation Co., Ltd., Qingdao 266100, China
3 School of Traffic and Transportation, Lanzhou Jiaotong University, Lanzhou 730070, China
* Correspondence: fuquanpan@yeah.net

**Abstract:** Aiming at the problem of metro operation and passenger transport organization under the impact of the novel coronavirus (COVID-19), a collaborative determination method of train planning and passenger flow control is proposed to reduce the train load rate in each section and decrease the risk of spreading COVID-19. The Fisher optimal division method is used to determine reasonable passenger flow control periods, and based on this, different flow control rates are adopted for each control period to reduce the difficulty of implementing flow control at stations. According to the actual operation and passenger flow changes, a mathematical optimization model is established. Epidemic prevention risk values (EPRVs) are defined based on the standing density criteria for trains to measure travel safety. The optimization objectives of the model are to minimize the EPRV of trains in each interval, the passenger waiting time and the operating cost of the corporation. The decision variables are the number of running trains during the study period and the flow control rate at each station. The original model is transformed into a single-objective model by the linear weighting of the target, and the model is solved by designing a particle swarm optimization and genetic algorithm (PSO-GA). The validity of the method and the model is verified by actual metro line data. The results of the case study show that when a line is in the moderate-risk area of COVID-19, two more trains should be added to the full-length and short-turn routes after optimization. Combined with the flow control measures for large passenger flow stations, the maximum train load rate is reduced by 35.18%, and the load rate of each section of trains is less than 70%, which meets the requirements of COVID-19 prevention and control. The method can provide a theoretical basis for related research on ensuring the safety of metro operation during COVID-19.

**Keywords:** metro; train plan; passenger flow control; particle swarm genetic algorithm; COVID-19; train load rate

## 1. Introduction

### 1.1. Background

With the spread and proliferation of the novel coronavirus (COVID-19), society as a whole has faced a broad-reaching and recurrent epidemic. To combat national epidemics, such as the COVID-19 outbreak, urban transportation should not only meet the requirements of dispatching emergency materials to medical treatment institutions in various disaster areas [1], but also meet the basic travel needs of urban residents and maintain the operation of cities. In the face of COVID-19, urban public transport (UPT) systems have the dual responsibility of ensuring effective transportation and hindering the spread of epidemics, but a region with great traffic accessibility means that the virus is also easy to reach its population [2–4]. The metro system plays an important role in the UPT system. However, due to its closed internal structure, dense passenger flow and strong mobility of people, it provides an extremely convenient environment for the spread of COVID-19 [5]. It

is important to study how to operate metro systems safely and efficiently under the impact of epidemic situations.

The spread of COVID-19 in the metro system is related to factors such as the dynamic changes in the number of passengers, the average commuting time, the one-way operating time of trains, and the condition of disinfection and ventilation. Among them, the passenger density in a carriage has a substantial impact on the number of possible infections at the time of an outbreak, and the lower the load level is, the fewer the number of possible infections [6]. Therefore, in the case that the spatial structure and ventilation conditions of the metro system cannot be considerably improved, the virus transmission pathway can be cut off, and the exposure rate can be reduced by wearing masks and controlling the number of people entering the station and riding on the train, thus effectively reducing the probability of the spread of airborne infections [7]. At the operational level, corporations should pay attention to the changes in passenger flow on the line network, optimize the organization of train transportation, and cooperate with measures such as passenger flow control and inbound epidemic prevention detection. By guiding passengers to travel rationally, the exposure time and the total number of exposed passengers in the possible transmission range of the virus are minimized, aggregated transmission is avoided, and public travel safety is ensured [8]. In March 2020, China's Beijing Metro Corporation changed its original train timetable, and the Changping Line and Batong Line began to adopt the special train timetable with high demands under unusual conditions, such as epidemics; that is, to operate by increasing the number of running trains and routes, reducing the departure interval and the number of stops, etc., with the aim of reducing the load rate of metro trains during COVID-19 [9] and gradually increasing the number of lines using the special train timetable to 13 in the following month. Based on the prevention and control idea of "source control-prevention-tracking-disposal", China's Wuhan metro operator designed a passenger flow control implementation plan during COVID-19, formulated quantitative indicators and assessed each link of passenger flow control individually in combination with qualitative analysis. After practical verification, all indicators meet the requirements of epidemic prevention, which can ensure a high level of operation service and has certain reference value for the operation of metros in other epidemic risk areas [10].

In the face of the impact of COVID-19, it is necessary to adjust the operation organization to reduce the load rate of metro trains, considering the interests of both passengers and corporations. At present, the existing epidemic prevention academic results of urban rail transit are all special operation diagrams designed for their own conditions by metro in COVID-19 risk places. Research on train plans and passenger flow control is mostly based on the normal environment, and there are few related studies on the impact of epidemic situations as a reference from theoretical aspects, such as models. Based on the above problems, this paper considers the requirements of metro COVID-19 prevention and control and proposes a collaborative determination method of metro train plan adjustment and passenger flow control, aiming to meet the travel demand while ensuring travel safety, improve the scientificity, accuracy and effectiveness of COVID-19 prevention, and provide a method reference for metro operation during the epidemic. The model and method are also applicable to other major public health events. To build the groundwork for the contributions of this paper, a literature review is provided in Section 1.2.

### 1.2. Related Literature

The global COVID-19 that broke out in 2020, and continues today, has disrupted normal production and life in many countries. In response to the impact of COVID-19 on the transportation industry, many scholars in the industry have conducted studies on issues in their fields. Hamidi et al. modeled the cumulative COVID-19 per capita infection rate in New York City using spatial lag models, found that crowding and average household size can greatly influence the spread of the virus, and suggested that policymakers should pay particular attention to neighborhoods with a high proportion of crowded households

and these destinations during the early stages of pandemics [11]. Zhou et al. developed a Susceptible-Exposed-Infectious-Removed (SEIR)-based model to simulate the spread of infectious diseases in the Tianyi metro station in Ningbo, China, performed sensitivity analysis on the parameters of the model and found that a higher contact rate, infectivity, and average illness duration were associated with higher numbers of infections [12]. Jin et al. constructed a PageRank algorithm-based risk model to accurately identify the key nodes of epidemic prevention and control in subway space, and the results of the case study showed that high-risk level spatial nodes are identified, including stairs, escalators, and platform transfers, which need to be focused on [13]. Seong et al. evaluated the correlations between subway use density and the activity of the influenza epidemic or COVID-19 pandemic using a time-series regression method [14]. Zhang et al. used a Bayesian network model to evaluate the operation risk of Chongqing Rail Transit in China under COVID-19 and constructed a risk assessment system based on the indicators of business risk, emergency management risk, passenger risk and public safety management risk [15]. Yang et al. studied the optimization of cold-chain emergency material distribution routes under COVID-19 and proposed a multidimensional robust optimization model to solve the problem of dispatching materials to medical treatment institutions in various disaster areas [16].

During COVID-19, by exploring the spatial and temporal distribution of passenger flow in metro networks, adjusting train operation plans, optimizing the allocation of line resources, and achieving accurate matching of capacity and traffic volume, a passenger flow control strategy is adopted to reduce the load of stations with high demand and better manage large passenger flow during peak hours to reduce the full load rate of the trains and ensure the safety of metro operation. For train operation plan optimization research, Deng et al. established a multiobjective bilevel programming model based on the passenger travel elastic demand function [17]. Zhang et al. analyzed the optimal train operation plan under different train types, train routing and stop modes [18]. Wang et al. established an optimization model with the objective of minimizing the total passenger waiting time and the transportation cost of corporations and with the constraints of the maximum number of available vehicles, train tracking interval and passenger demand [19]. Liu et al. established a model of long and short routing operation plans based on the cycle analysis method of train working diagrams, with the goal of minimizing the total cost converted from each cost. The actual case shows that the method has a good optimization effect on the radius line [20]. Zhang et al. developed a mixed integer linear programming (MILP) model for train scheduling, where short turning train services and full-length train services are optimized based on the predefined headway obtained by passenger demand analysis [21]. Li et al. proposed a nonuniform departure method to balance the load of full-length and short turning trains and established a mathematical model with the goal of minimizing passenger waiting time and corporation operating costs. A two-stage genetic algorithm was designed to solve the problem [22]. Li et al. established an optimization model for multirouting and multigrouping metro lines and generated alternative routing sets through the location of turn-back stations. Considering the factors that passengers choose to take trains with different routings, they optimized the train routing, marshalling number and operation logarithm in the train diagram [23]. Ma et al. constructed an emergency customized bus route optimization method considering epidemic prevention and control requirements under public health emergencies, and the number of operating vehicles and operating hours were increased after optimization, which is useful for metro operations [24].

In passenger flow control research, Jiang et al. studied the congestion of metro platform stairs during peak hours and developed specific measures to control passenger flow and ease congestion [25]. Lu et al. predicted origin-destination (OD) passenger flow based on the idea of flow on the path so that it can play a good role in passenger flow warning [26]. Xu et al. analyzed passenger flow line metro stations, established a path selection model, and carried out corresponding control measures for passenger flow based

on the threshold of facilities and equipment [27]. Guo et al. analyzed the inflow passenger flow of public transport stations with capacity constraints and adopted coordinated control measures among stations [28]. Li et al. considered the timetable and passenger flow control strategy of high traffic density metro lines and carried out the optimal control of passengers according to the characteristics of the timetable [29]. Xu et al. analyzed the dynamic change process of inbound and transfer passenger flows, established a multistation cooperative passenger flow control model, and used a logit model to describe the route choice behavior of passengers [30]. Xue et al. divided stations into four areas: platform, paid zone, nonpaid zone, and station entrance. According to the train capacity and number of people stuck on a platform, a three-level passenger flow control model was established to solve the optimal strategy for controlling the number of people in each area [31]. Liu et al. proposed a passenger flow control strategy based on queuing theory and simulated the congestion propagation between service facilities in a station through Simulink. The optimization goal is to minimize passenger delay time and corporation operating costs [32].

It is important to note that in urban rail transit passenger transportation organization problems, simply optimizing the train plan to meet passenger demand will encounter a bottleneck when the capacity reaches the upper limit, and simply taking a passenger flow control strategy to accommodate the transport capacity will reduce the service level to passengers. At the same time, metro operation under the impact of COVID-19 is still a new research field, and systematic research focusing on safety prevention and control in the context of COVID-19 is not yet complete. Therefore, this paper constructs a collaborative optimization model of train plans and passenger flow control under the impact of COVID-19, fully considers the influence of COVID-19 prevention and control in the objective function of the model, and studies the integrated method of train plans and passenger flow control, which is of great significance to metro operation organization under the influence of public health emergencies.

The structure of the rest of this paper is as follows: Section 2 describes the metro operation organization problem from the two aspects of the train plan and passenger flow control. Section 3 constructs metro train plan adjustment and passenger flow control mathematical models to reduce the risk of epidemic transmission. Section 4 introduces the method of model processing and the algorithm design of model solving. Section 5 presents the case analysis. Section 6 provides a discussion of the study. Section 7 is the conclusion.

## 2. Problem Description

The purpose of adjusting the train plan and implementing passenger flow control measures is to reduce the load rate of trains to meet the requirements of COVID-19 prevention and control. Figure 1 shows the idea of the method.

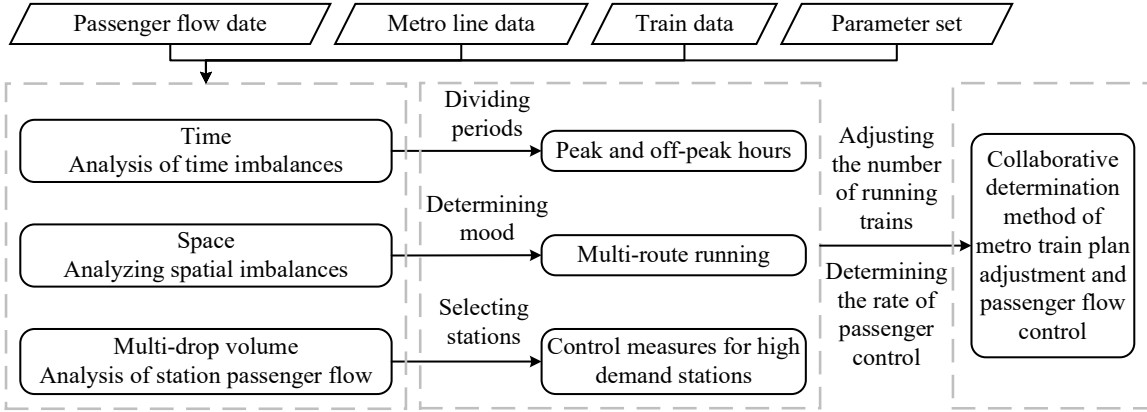

**Figure 1.** Basic process for designing the method in epidemics.

In the adjustment of the train plan, by analyzing the spatial and temporal characteristics of the passenger flow distribution, according to the unbalanced characteristics of

passenger flow in the time dimension, the peak and off-peak hours are divided, and the routing plan is determined according to the spatial imbalance characteristics. Specifically, the time imbalance coefficient of passenger flow is calculated according to the passenger flow information, and the period when the time imbalance coefficient is greater than 1.5 is set as peak hours. Then, the spatial imbalance coefficient of passenger flow in peak hours is calculated. If the spatial imbalance coefficient of each interval is less than 1.5, the full-length train plan is adopted; otherwise, the full-length and short turning train plan is adopted, the short turning route contains all intervals where the spatial imbalance coefficient is greater than 1.5, and the intervals where the coefficient is greater than 1 are included as much as possible.

In the passenger flow control strategy, the main commuter stations in peak hours are managed. Due to the relatively fixed starting and ending points of companies, schools and homes, it can be considered that the OD ratio between line stations remains unchanged during this period. It can be estimated by the Automatic Fare Collection (AFC) data, and combined with the curve of the arrival rate changing with time shown in Figure 2, the number of OD demands between lines and stations in a certain period of time can be further calculated [33]. The expressions are given in Equations (1) and (2):

$$\Phi_{s_1}(\tau) = \int_0^\tau \lambda_{s_1}(t)dt \tag{1}$$

$$\Phi_{s_1,s_2}(\tau) = \Phi_{s_1}(\tau) \cdot e_{s_1,s_2} \tag{2}$$

where $\Phi_{s_1}(\tau)$ is the cumulative arrival passenger flow of station $s_1$ at time $\tau$; $\lambda_{s_1}(t)$ is the arrival rate at station $s_1$; $\Phi_{s_1,s_2}(\tau)$ is the cumulative passenger demand from station $s_1$ to station $s_2$ at time $\tau$; and $e_{s_1,s_2}$ is the proportion of all passengers departing from station $s_1$ to station $s_2$.

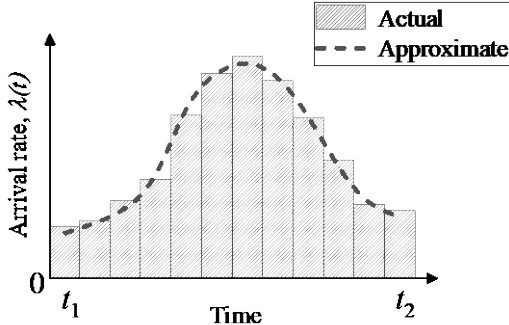

**Figure 2.** Curves of passenger flow arrival.

To simplify the problem, this paper only implements the passenger flow control strategy for one metro line direction, and the other direction can be operated according to the same principle. In the existing research on metro passenger flow control, the optimization scheme given for the example is mostly the flow control rate or the number of people allowed to enter the station in each departure interval or each unit time period during the research period [31,34,35]. In actual implementation, the flow control strategy needs to be changed frequently. To reduce the difficulty of passenger transport organization at stations, the study period can be divided into several statistical periods, and the time series and matrix of inbound passenger flow in Equations (3) and (4) are constructed. The Fisher optimal division method is used to cluster the statistical periods with similar passenger flow characteristics in AFC data to obtain passenger flow control periods [36]. The Fisher optimal division method is a clustering method in statistics, and its principles are not described here.

$$H = \{H_1, H_2, \ldots, H_r\} \tag{3}$$

$$H_r = [p_1^r, p_2^r, \ldots, p_s^r]^T \tag{4}$$

where $H$ is the arriving passenger flow time series; $H_r$ is the arriving passenger flow matrix in the $r$th statistical period; and $p_s^r$ is the number of arriving passengers at station $s$ during the $r$th statistical period.

## 3. Model Construction

### 3.1. Assumptions, Notions and Decision Variables

The following assumptions are made to construct the model for the studied problem:

(1)    The OD ratio between stations remained unchanged during the study period. The origin and destination points of companies, schools and homes are relatively fixed, and the OD structure of the lines during this time period can be considered to be relatively stable, i.e., passengers will not choose to take other modes of transportation or change their choice of departure and destination stations due to the adoption of passenger flow control measures.

(2)    The train departure interval, interval running duration and stopping duration are in accordance with the pre-determined timetable, without considering the occurrence of late and delay due to unexpected events. In actual operation, the punctuality of urban rail transit trains is extremely high in the absence of breakdowns, which can be assumed to operate strictly according to the train timetable.

(3)    Simplifying the process of passenger entry and exit. The model considers the transportation organization optimization strategy mainly at the line level and therefore ignores the influence of the station infrastructure, i.e., passengers passing through the inbound entrance can reach the station platform directly after the restricted flow, and since the exiting passengers eventually leave the metro station, which can reduce metro passenger congestion, it is assumed that all exiting passengers can leave the rail station smoothly without accumulating in the station.

Let the set of all stations of the metro line be $S = \{1, 2, \ldots, n\}$, and the indices are $s$ and $u$. The turn-back stations of the short turning train indices are $s_0$ and $s_1$. The intervals index is $s\prime$, $s\prime$ and $s$ have a one-to-one correspondence, and $s\prime \neq n$. The training set is $I$, and the index is $i$. The train routing set is $Y = \{1, 2\}$, where 1 represents the full-length route and 2 represents the short turning route. The control period set is $K$, and the index is $k$.

The decision variables of the model are as follows: $J_y$ represents the number of trains running on each route; $\mu_{s,k}$ represents the flow control rate of stations implementing passenger flow control in each control period.

### 3.2. Analysis of Passenger Dynamic Change

The dynamic change process of passengers in the metro system mainly includes entering the station, boarding the train, riding between intervals and getting off the train. For station $s$ within the control period $k$, train $i$ passing through the station and the previous train $i\prime$, the calculation expressions of the number of passengers entering the station during the departure interval are as follows:

$$q_{is}^{In} = Q_{is} \cdot (1 - \mu_{is}) \tag{5}$$

$$\mu_{is} = \mu_{s,k}, \ \tau_{i\prime s}^{D}, \tau_{is}^{D} \in k \tag{6}$$

where $q_{is}^{In}$ is the actual number of passengers entering station $s$ during the departure interval between train $i$ and previous train $i\prime$; $Q_{is}$ is the amount of inbound demand at station $s$ during the departure interval between train $i$ and previous train $i\prime$; $\tau_{is}^{D}$ is the time when train $i$ leaves station $s$; and $\mu_{is}$ is the flow control rate of station $s$ during the departure interval between train $i$ and previous train $i\prime$. If $\tau_{i\prime s}^{D} \in k$, $\tau_{is}^{D} \in k+1$, then $\mu_{is}$ is calculated by linear weighting.

The inbound passenger flow demand includes new arrivals and passengers who fail to enter the station due to passenger flow control in the previous departure interval. The calculation expression is as follows:

$$Q_{is} = q_{is} + x_{(i-1)s} \tag{7}$$

where $q_{is}$ is the number of arriving passengers at station $s$ during the departure interval between train $i$ and previous train $i\prime$; $x_{is}$ is the number of people stranded at station $s$ during the departure interval between train $i$ and previous train $i\prime$, and $x_{1s} = 0, x_{is} = Q_{is} - q_{is}^{In}$.

We calculate the number of passengers arriving at station $s$ within the corresponding departure interval according to Equations (1) and (2). The expression is as follows:

$$q_{is} = \begin{cases} \Phi_s(\tau_{is}^D) - \Phi_s(\tau_{i\prime s}^D), & 1 \leq s < n \\ 0, & s = n \end{cases} \tag{8}$$

When the train plan is full-length and short turning routing, for the short turning routing section, when train $i$ passes through station $s$, the number of passengers boarding the train from the platform is the same as the number of passengers entering the station during the departure interval between train $i$ and previous train $i\prime$, while the passengers in the short turning routing section choose to take the routing train according to the destination, so the calculation expressions of the number of passengers boarding the train are as follows: or

$$q_{is}^B = \begin{cases} \sum\limits_{u=s+1}^{n} e_{s,u} \cdot q_{is}^{In}, & 1 \leq s < s_0 \text{ or } s_1 \leq s \leq n \\ \sum\limits_{u=s+1}^{n} e_{s,u} \cdot q_{is}^{In} \cdot \gamma_i, & s_0 \leq s < s_1 \end{cases} \tag{9}$$

$$\gamma_i = \begin{cases} 1, & i \in I_1 \\ \left( \sum\limits_{u=s+1}^{s_1} e_{s,u} \right) \Big/ \left( \sum\limits_{u=s+1}^{n} e_{s,u} \right), & i \in I_2 \end{cases} \tag{10}$$

where $q_{is}^B$ is the number of passengers boarding train $i$ at station $s$; $\gamma_i$ is the proportion of passengers choosing different routing trains; and $I_1$ and $I_2$ are the train sets of full-length routing and short turning routing, respectively.

The number of passengers deboarding at station $s$ is the number of passengers whose destination is station $s$ among all the passengers boarding at the station ahead, and the calculation expression is as follows:

$$q_{is}^L = \begin{cases} 0, & s = 1 \\ \sum\limits_{u=1}^{s-1} q_{is}^B \cdot e_{u,s}, & 1 < s \leq n \end{cases} \tag{11}$$

where $q_{is}^L$ is the number of passengers leaving train $i$ at station $s$.

The number of passengers carried during the interval operation of the train can be calculated by the number of boarding and alighting passengers, and the expression is as follows:

$$c_{is\prime} = \begin{cases} q_{is}^B, & s\prime = 1 \\ c_{i(s\prime-1)} - q_{is}^L + q_{is}^B, & 1 < s\prime \leq n \end{cases} \tag{12}$$

where $c_{is\prime}$ is the number of passengers carried by train $i$ in interval $s\prime$.

### 3.3. Objective Function

(1)　Minimizing Epidemic Prevention Risk Values

According to Cao et al. [37] and Yu et al. [38], COVID-19 patients mainly have related symptoms during the onset period, and there are some asymptomatic cases. Considering the common situation, the model in this paper assumes that a patient does not cough and sneeze during the ride and only maintains normal breathing, and the effective transmission distance of the virus is 1 m. For the ticket office and the queue of passenger flow control in a metro station, the safe distance between passengers can be ensured by means of management guidance and reserved queuing space. For the train compartment, according to the relevant criteria of China's existing train standing density, when the number of

people in carriages reaches the personnel quota (six people per square meter), passengers' perception is in a critical loose and crowded state, and beyond this state, it cannot guarantee safe social distancing under COVID-19. Therefore, the epidemic prevention risk value (EPRV) is defined to measure the safety of passenger travel under the impact of COVID-19. When there is no one standing in a carriage and all passengers have seats, the EPRV is 0, indicating that passenger safety is fully guaranteed, and the probability of COVID-19 spreading in the carriage is minimal. Additionally, when the standing density is greater than six people per square meter, the EPRV is 1, indicating a complete failure to meet the requirements of COVID-19 prevention and control, and passengers traveling on the metro cannot be guaranteed to be safe. The specific calculation expression of EPRV is as follows:

$$R_{is\prime} = \begin{cases} 0, & c_{is\prime} \leq bC_0 \\ \frac{c_{is\prime} - bC_0}{bC - bC_0} & bC_0 < c_{is\prime} \leq bC \\ 1, & c_{is\prime} > bC \end{cases} \tag{13}$$

where $R_{is\prime}$ is the EPRV of train $i$ in interval $s\prime$; $b$ is the number of train formations; $C_0$ is the number of seats in carriages; and $C$ is the personnel quota of the train.

To reduce the possibility of COVID-19 spread, the EPRV of trains running in each interval of the line should be reduced as much as possible. The objective function is as follows:

$$\min Z_1 = \sum_{i \in I} \sum_{s\prime=1}^{n-1} R_{is\prime} / (n-1) \tag{14}$$

where $Z_1$ is the average risk level of COVID-19 spread in intervals.

(2)    Minimizing passenger waiting time

The passenger waiting time includes the waiting time for the train and the detention time at the passenger flow control place, which is calculated separately for full-length and short turning route sections. The expressions of the objective function are as follows:

$$Z_{21} = \sum_{i \in I_1} \left[ \sum_{s=1}^{s_0-1} \left( x_{is} \cdot \Delta t_1 + q_{is}^{In} \cdot \frac{\Delta t_1}{2} \right) + \sum_{s=s_1+1}^{n} \left( x_{is} \cdot \Delta t_1 + q_{is}^{In} \cdot \frac{\Delta t_1}{2} \right) \right] \tag{15}$$

$$Z_{22} = \sum_{i \in I_2} \left[ \sum_{s=s_0}^{s_1} \left( x_{is} \cdot \Delta t_2 + q_{is}^{In} \cdot \frac{\Delta t_2}{2} \right) \right] \tag{16}$$

$$\min Z_2 = Z_{21} + Z_{22} \tag{17}$$

where $Z_{21}$ and $Z_{22}$ are the passenger waiting times for full-length route and short turning route sections, respectively; $\Delta t_1$ and $\Delta t_2$ are departure intervals for one full-length route and short turning route sections, respectively; and $Z_2$ is the total waiting time for passengers.

(3)    Minimizing corporation operating costs

Corporation operating costs include train running costs and labor costs, with the train running kilometers and operating time. The objective function expression is as follows:

$$\min Z_3 = g \cdot \sum_{y=1}^{2} J_y \cdot l_y + h \cdot \sum_{y=1}^{2} J_y \cdot \theta_y \tag{18}$$

where $Z_3$ is the operating cost of the corporation; $g$ is the unit cost of train running kilometers; $l_y$ is the length of train routes; $h$ is the unit time cost of labor; and $\theta_y$ is the operation time of train routes.

### 3.4. Constraints

The constraints of the collaborative optimization model of metro train plan adjustment and passenger flow control are expressed as follows:

$$\mu_{is} \leq M \tag{19}$$

$$q_{is}^B \leq \min\left\{ w_{is}^{(1)}, w_{is}^{(2)}, w_{is}^{(3)} \right\} \tag{20}$$

$$q_{is}^{In} \leq W_i \tag{21}$$

$$c_{is\prime} \leq \psi \cdot b \cdot C \tag{22}$$

$$\frac{P_{s\prime}}{\sum\limits_{y \in Y} \varphi_{s\prime,y} J_y b C} \leq \alpha_0 \tag{23}$$

$$\tau_{is}^A - \tau_{i\prime s}^A \geq \Delta t_{\min} \tag{24}$$

$$\tau_{is}^D - \tau_{i\prime s}^D \leq \Delta t_{\max} \tag{25}$$

$$a \cdot J_1 = J_2, a \in N \tag{26}$$

where $M$ is the maximum flow control rate; $w_{is}^{(1)}$, $w_{is}^{(2)}$ and $w_{is}^{(3)}$ are the maximum capacity of the gates, escalators and horizontal channels of station $s$ during the departure interval between train $i$ and previous train $i\prime$, respectively; $W_i$ is the maximum capacity of station $i\prime$s platform; $\psi$ is the maximum overload factor of the train; $P_{s\prime}$ is the section passenger volume of interval $s\prime$; $\varphi_{s\prime,y}$ is a 0–1 binary variable; if interval $s\prime$ belongs to the range of routing $y$, then $\varphi_{s\prime,y} = 1$; otherwise $\varphi_{s\prime,y} = 0$; $\alpha_0$ is the maximum load rate of epidemic prevention requirements according to the standard of metro operation in China during COVID-19; $\alpha_0$ for low-, medium- and high-risk areas of COVID-19 are not a requirement and are 70% and 50%, respectively; $\tau_{is}^A$ is the time when train $i$ arrives at station $s$; $\Delta t_{\min}$ is the train tracking interval; $\Delta t_{\max}$ is the maximum departure interval; and $N$ is the set of natural numbers.

Equation (19) is the maximum passenger flow control rate constraint; Equation (20) is the capacity constraint of facilities and equipment in the station; Equation (21) is the capacity constraint of station platform; Equation (22) is the train carrying capacity constraint; Equation (23) is the load rate constraint considering the requirements of COVID-19 prevention; Equation (24) is the tracking train interval constraint; Equation (25) is the maximum departure interval constraint; and Equation (26) is the balance constraint of full-length and short turning route running trains.

## 4. Model Solving

### 4.1. Transformation of the Model

The model proposed in Section 3 is a multiobjective constrained optimization model. The solution idea is to transform the original model into a single-objective model. The linear weighted sum method is used to transform the three objectives of the EPRV, passenger waiting time and corporation operating cost into a single objective. The expressions are as follows:

$$\min Z = \omega_1 Z_1 + \omega_2 Z_2 + \omega_3 Z_3 \tag{27}$$

$$\omega_1 + \omega_2 + \omega_3 = 1 \tag{28}$$

where $\omega_1$, $\omega_2$ and $\omega_3$ are weighting factors.

Due to the different dimensions of the objectives, the objective values under the existing operating plan (Let them be $Z_1\prime$, $Z_2\prime$ and $Z_3\prime$) are first nondimensionalized as benchmark values, that is, let $\omega_1 Z_1\prime = \omega_2 Z_2\prime = \omega_3 Z_3\prime$, and take $\omega_1 = 1$, so $\omega_2 = \omega_1 Z_1\prime / Z_2\prime$, $\omega_3 = \omega_1 Z_1\prime / Z_3\prime$. The weighting coefficients are then normalized, that is, $\omega_1 = \omega_1 \Big/ \sum\limits_{\delta=1}^{3} \omega_\delta$, $\omega_2 = \omega_2 \Big/ \sum\limits_{\delta=1}^{3} \omega_\delta$, $\omega_3 = \omega_3 \Big/ \sum\limits_{\delta=1}^{3} \omega_\delta$.

### 4.2. Algorithm Design

Because of the NP (nondeterministic polynomial) characteristics of the problem and model studied in this paper, a hybrid optimization algorithm based on particle swarm optimization and a genetic algorithm (PSO-GA) is designed, which combines the simple

operation and ease of implementation of PSO with the strong global search capabilities and fast convergence of the GA to better solve the model. The solution process is shown in Figure 3.

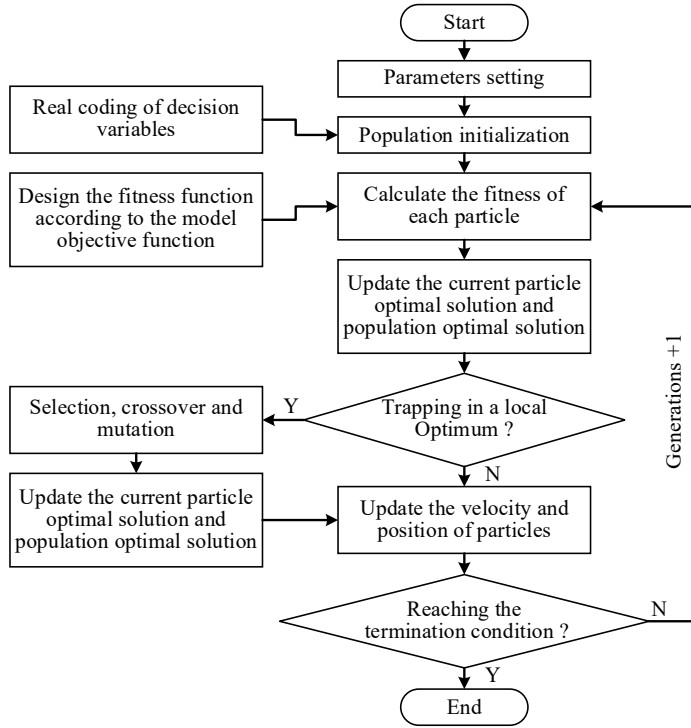

**Figure 3.** Flow chart of the solution algorithm.

(1) The real-number encoded forms of all decision variables are used as the initial positions of the particle. A total of *m* feasible particles are randomly generated as the initial population, and each particle is randomly assigned an initial speed within a certain range. The maximum number of iterations is denoted by *G*.

(2) According to the objective function of the model, the fitness function of the algorithm is designed, the fitness function value of each particle is calculated, and the optimal solution of the current particle and the optimal solution of the population are recorded.

(3) The velocity and position of the particle are updated according to the inertia weight $\delta$ and the learning factor $\eta_1, \eta_2$ parameters.

(4) If the algorithm is trapped in a local optimum (the optimal solution remains unchanged for a certain number of iterations), the operations of selection, crossover and mutation in the genetic algorithm are performed, with crossover and mutation probabilities, $\varepsilon_c$ and $\varepsilon_v$, in the manner of a simulated binary crossover and polynomial mutation; if it is not trapped in a local optimum, then the next step is carried out.

(5) We determine whether the maximum number of iterations *G* is reached. If it is reached, we terminate the algorithm and output the result; otherwise, we continue the iteration.

## 5. Case Study

### 5.1. Case Description and Parameter Setting

Based on the relevant data of Qingdao Metro Line 1 in Shandong Province, China, a case study is conducted to verify the feasibility and effectiveness of the proposed method and model. Line 1 is a north–south trunk line connecting downtown and distant residential areas with 39 stations, and stations 1, 6, 20, 28, and 39 have turnback conditions. The line information is shown in Figure 4. The study period is 18:00–19:00 on a weekday evening peak (the time imbalance coefficient is greater than 1.5). During this period, the up direction is the direction of heavy passenger flow. The spatial imbalance coefficient of passenger flow is shown in Figure 5, and the arriving passenger flow of each station is shown in Figure 6.

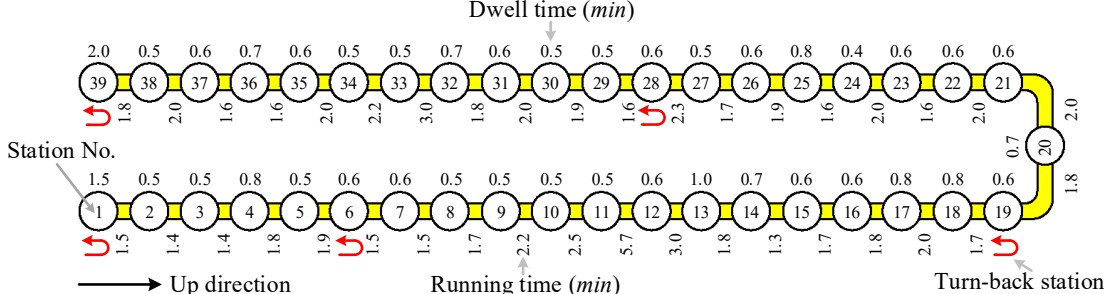

**Figure 4.** Qingdao Metro Line 1.

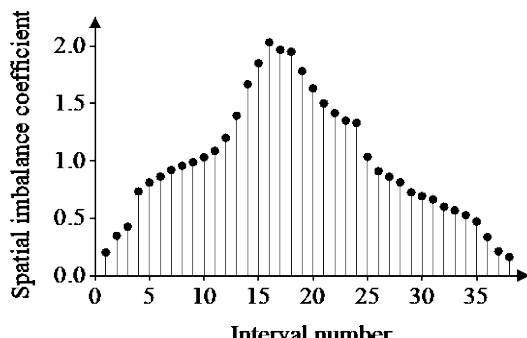

**Figure 5.** Spatial imbalance in passenger flows.

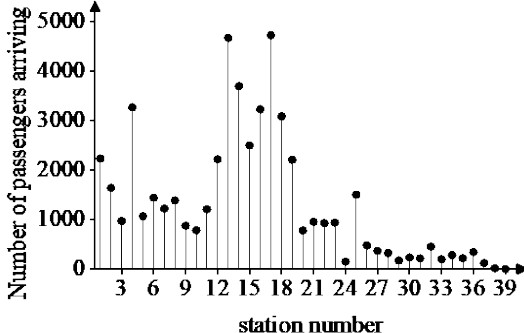

**Figure 6.** Passenger arrivals at stations.

In Figure 5, it can be seen that the spatial imbalance coefficient of 8 sections in this direction is greater than 1.5, so it is necessary to carry out full-length and short turning routes. Combined with the actual operation situation, the short turning route turn-back station is determined as stations 6 and 28, which is consistent with the existing scheme. According to Figure 6 and the situation of the line station, passenger flow control measures are taken for the high-demand stations (4, 13, 14, 16, 17 and 18).

According to the information provided by Qingdao Metro Group Co., Ltd., (Shanghai, China) the parameters in the proposed model are as follows: $\Delta t_{min} = 120$ s, $\Delta t_{max} = 450$ s, $C = 1440$ person, $g = 30$ Y/km, $h = 15$ Y/min, $M = 0.5$, and $\psi = 1.2$. During COVID-19, people will choose transportation vehicles based on personal attributes, travel attributes, and perception of COVID-19 and pay more attention to the safety of public transportation such as metro [39].

Under the impact of COVID-19, the influencing factors of metro passenger transport organization still mainly consider passenger demand, train operation conditions and capacity resources, but the specific content focuses more on travel safety. In this paper, the importance of safety, promptness, punctuality and economy during COVID-19 is analyzed through questionnaire surveys of passengers and managers, and the respective weight coefficients are determined by matching the importance of the three subobjectives in the

objective function under different risk levels of COVID-19. The specific values and their changing trends are shown in Figure 7.

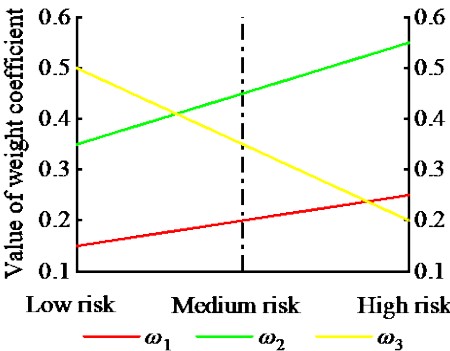

**Figure 7.** Value and trend of the weight coefficients.

The algorithm solution process is performed using MATLAB (version R2021a, Natick, MA, USA) on a personal computer, and the parameters are set as follows: $m = 100$, $G = 200$, $\delta \in [0.2, 0.9]$, $\eta_1 = \eta_2 = 2.0$, $\varepsilon_c = 0.9$, and $\varepsilon_v = 0.2$.

*5.2. Results and Analysis*

The experiment assumes that the metro line is in a medium-risk area of COVID-19. According to the content described in Section 1, the research period is divided into 12 statistical periods with 5 min as the time granularity extraction, and the time series and matrix of inbound passenger flow are constructed. The change trend of the classification loss function is depicted in Figure 8.

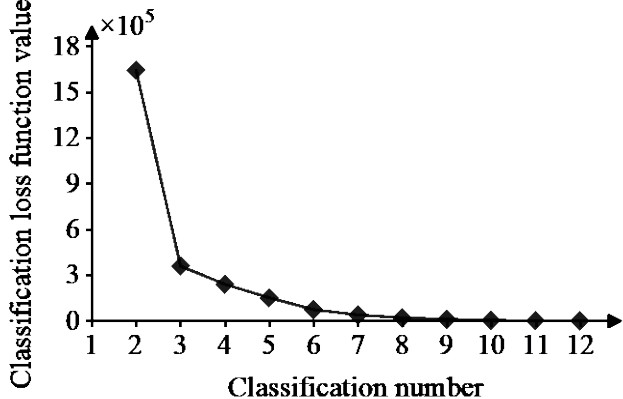

**Figure 8.** Diagram of the classification loss function.

Figure 8 shows that when the classification number is 3, the inflection point of the curve is obvious. According to the principle of the Fisher optimal division method, all statistical periods can be divided into three categories. The results are shown in Table 1.

**Table 1.** Determination of control periods.

| Category Number | Statistical Periods | Passenger Flow Control Periods |
| :---: | :---: | :---: |
| 1 | No.1~No.4 | 18:00–18:20 |
| 2 | No.5~No.8 | 18:20–18:40 |
| 3 | No.9~No.12 | 18:40–19:00 |

According to the results of the flow control period division, the relevant parameters are brought into the model for solving. Figure 9 shows the solution process of the algorithm.

The target value of the optimal individual exhibits a downward trend before the 55th iteration and stabilizes and reaches the optimal solution after the 55th iteration.

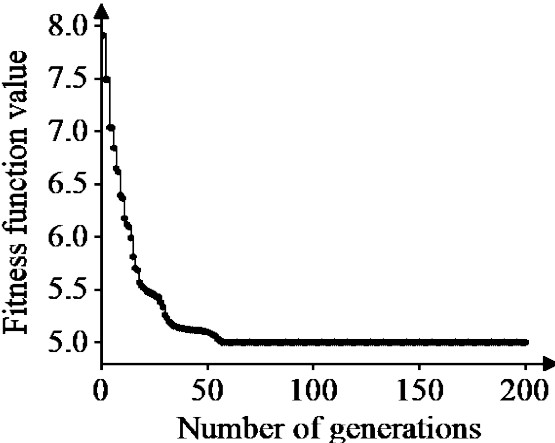

**Figure 9.** Algorithm solution process.

The experimental results are as follows: the number of trains running on both full-length and short turning routes is 10, which is two more than the existing scheme, and the transportation capacity of the short turning is increased by 25%. The flow control rate of the experimental station in each passenger flow control period is shown in Table 2. The passenger waiting time on the whole line is 1942.5 h, and the operating cost is CNY 53,208.3, which are 0.79% and 24.7% higher than the existing scheme, respectively. The load rate $\alpha$ of trains in the short turning route section before and after optimization is shown in Figure 10 (each line represents the running track of every train, the full-length and short turning route trains run alternately, and the first running track is the full-length route train).

**Table 2.** Passenger flow control rate at stations.

| Control Periods | Station Number | | | | | |
|---|---|---|---|---|---|---|
| | 4 | 13 | 14 | 16 | 17 | 18 |
| 1/% | 42 | 26 | 22 | 13 | 20 | — |
| 2/% | 28 | 31 | 30 | 22 | 27 | 17 |
| 3/% | 13 | 19 | 18 | — | 15 | — |

Ps: —indicates no passenger flow control measures.

In Figure 10a, it can be seen that the crowded areas are concentrated from Station 11 to Station 24 during the evening peak period, especially some sections between Station 16 and Station 19, where colors are almost all dark red, indicating that the load rate of the train is very high, the train is extremely congested and COVID-19 in the compartment spreads easily. The optimization results in Figure 10b show that under the condition that the passenger waiting time and the total operating cost of the whole line have not increased substantially, the full load rate of each train running during this period obviously decreases, 105 red sections and 22 dark red sections have disappeared, the maximum load rate is reduced by 35.18%, and the load rate of each section meets the requirements of COVID-19 prevention and control of no more than 70%. There are two reasons for the decrease in the maximum load rate of the optimized train. First, the increase in the number of running trains and the capacity provided by the operation of the full line optimize the allocation of resources on the supply side. Second, the implementation of flow control measures slows the concentration of passenger flow into the station to a certain extent, and the amount of passenger flow into the station in the whole period is more balanced, avoiding excessive demand in a certain period of time.

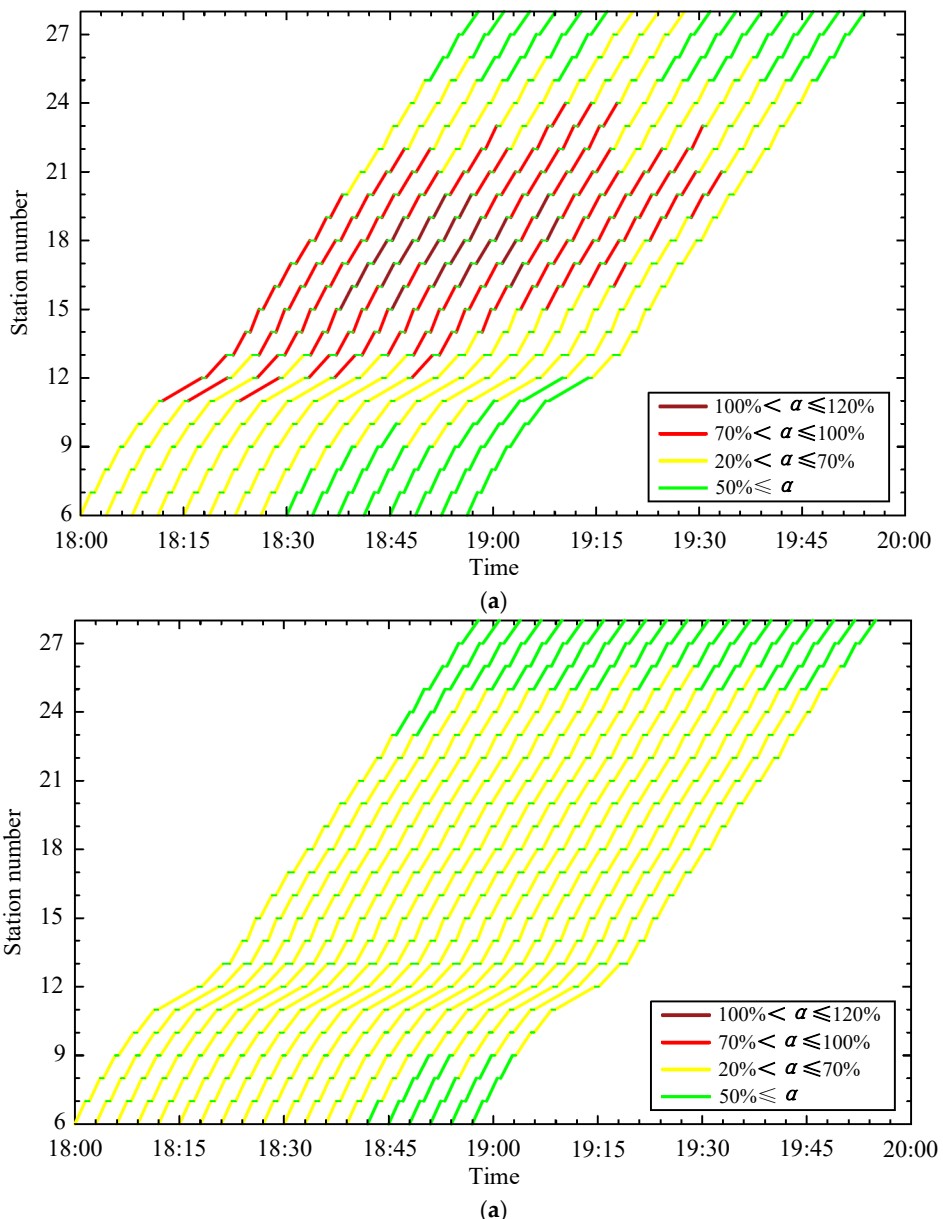

**Figure 10.** Load rate of the trains in the short turning route. (**a**) Before optimization; (**b**) after optimization.

Figure 11 shows the number of arriving passengers, inbound passengers and detained passengers in each control period at the experimental stations. Most of the passengers at Station 4 arrived in the first 20 min, and the rest of the stations arrived in 20–40 min. When passenger flow control measures are taken, the peak difference in the number of arriving passengers is less than the peak difference in the number of inbound passengers, indicating that the inbound passenger flow is gentler, avoiding excessive passenger flow density in a short period of time and reducing the risk of COVID-19 transmission. At the same time, there is a certain degree of passenger flow gathering at each experimental station. The maximum number of detained passengers at Station 4 is approximately 400, and the maximum number of detained passengers at stations 13, 14 and 17 can reach approximately 300. The number of detained passengers is related to the overall demand of the station and the intensity of flow control. In actual operation, sufficient space should be reserved for the gathering area of detained passenger flow, the frequency and guidance of staff inspection should be strengthened, and measures, such as maintaining the queuing

interval and inbound COVID-19 prevention detection, should be taken to further reduce the probability of COVID-19 transmission and to ensure travel safety.

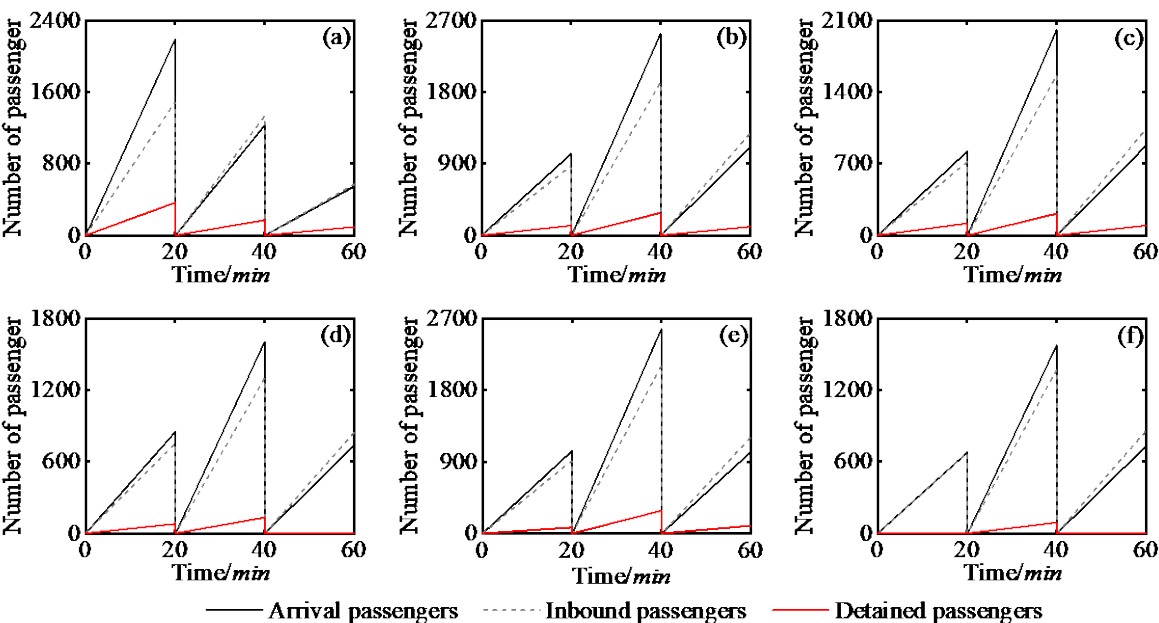

**Figure 11.** Number of arriving passengers, inbound passengers and detained passengers at control stations: (**a**) station 4; (**b**) station 13; (**c**) station 14; (**d**) station 16; (**e**) station 17; (**f**) station 18.

## 6. Discussion

(1) In the modeling process of the collaborative determination method of metro train plan adjustment and passenger flow control, the train stop plan, the turn-back mode and the change process of passenger walking are simplified. In future research, other factors can be considered comprehensively on the basis of this method, and more in-depth research can be carried out to further improve the train diagram design and the implementation method of passenger flow control.

(2) During the impact of COVID-19, especially when a line passes through high-risk areas of COVID-19, some passengers may take measures such as home isolation or home offices, which will change the passenger flow, structure and OD of the line. The specific degree of change and the impact on metro passenger transport organizations need further research.

(3) The specific train plan and passenger flow control measure also need to be combined with the capacity resources of the metro operation corporation, layout of the equipment and facilities of each station, power system, artificial factors and other conditions. This paper establishes a mathematical model of the coordinated implementation of train plan adjustment and passenger flow control under the influence of COVID-19 and combines the design of PSO-GA to obtain the optimization plan, which may have a certain gap with the actual operation situation.

## 7. Conclusions

(1) Considering the problem of metro passenger transportation and organization under the influence of COVID-19, a collaborative determination method of metro train plan adjustment and passenger flow control is proposed. According to the actual operation and the dynamic change in passenger flow, a mathematical optimization model is established. The epidemic prevention risk value is defined to measure the safety degree of the train in each interval. The minimum epidemic prevention risk value is taken as one of the optimization objectives of the model. The other two optimization objectives are the minimum passenger waiting time and the minimum operating cost of the corporation.

(2)　The research period division scheme is formulated by the Fisher optimal division method, and the results are used as the basis for solving the model. PSO-GA is designed to solve the model. The chromosome crossover and mutation operations in the genetic algorithm are integrated into the iterative process of particle swarm optimization to improve the solution efficiency.

(3)　The validity of the model and algorithm is verified by the case of Qingdao Metro Line 1 in China. The results show that when the line is in the risk area of COVID-19, two trains with full-length and short turning routes should be added after optimization, and the stations with high passenger demand adopt different flow control rates in various control periods. The combination of the two can alter the load rate of trains in each interval by less than 70%, effectively reduce the personnel density in carriages, reduce the risk of COVID-19 spread and cluster infection, and control the increase in passenger waiting time and enterprise operation cost in the whole line at an acceptable level.

**Author Contributions:** Methodology, J.L.; Data curation, H.T.; Writing—original draft, J.L.; Supervision, F.P., C.M., L.Z. and X.Y. All authors have read and agreed to the published version of the manuscript.

**Funding:** This work is supported by the Shandong Provincial Natural Science Foundation of China (ZR2020MG021), the Humanities and Social Science Planning Fund of Chinese Ministry of Education (18YJAZH067), and the Natural Science Foundation of China (62003182).

**Institutional Review Board Statement:** Not applicable.

**Informed Consent Statement:** Not applicable.

**Data Availability Statement:** Not applicable.

**Conflicts of Interest:** The authors declare no conflict of interest.

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
