# Peer review of "Collaborative Determination Method of Metro Train Plan Adjustment and Passenger Flow Control under the Impact of COVID-19"

_sustainability, doi:10.3390/su15021128_

Round 1
Reviewer 1 Report
You did not put references in parentheses or anything.
2- carriage is not a common word in transportation, it reminds me of the horse.
3- introduction and literature review are not separated. So you should divide them first, then summarize the whole literature in a table and mention the gap in the recent research.
4- When considering the macro analysis, your hypothesis is incorrect
5- Section3, there are many assumptions that do not make any sense. You did not conder the impedance for the passengers
6- what is the relevancy between waiting time and risk level of covid? we can still spread covid when we are on the train.
7- w1 and w2 and w3 have the same weight? if not, elaborate more.
Reviewer 2 Report
Major comments:
I find the manuscript “Collaborative determination method of metro train plan adjustment and passenger flow control under the impact of COVID-19” is an interesting study to discuss the metro operation with the prevention of COVID-19. However, I am reluctant to accept the manuscript in the current form and I suggest a major revision needs to be performed.
The major drawback of the manuscript is the way evaluate the energy performance certification system. But the 11 relevant parameters in table 6 are used to describe the energy consumption level of a typical underground station were from 8 different literatures, without any revision or credible description of the consequences of mixing these parameters. Given the fact that Authors rely heavily upon these parameters I insist that they do some more systematic analysis of the presented works.
There are other problems shall be improved.
1) It is noted that the manuscript needs careful editing by someone with expertise in technical English editing paying particular attention to English grammar, spelling, and sentence structure so that the goals and results of the study are clear to the reader.

Reviewer 3 Report
1. References should be cited in square brackets.
2. Assumption no 3 (line 193) - and if there are queues, they won't leave quickly. Are queues analyzed in terms of waiting time and queue length?
Round 2
Reviewer 2 Report
This manuscript has been much improved.
